# Nanocellulose from Cotton Waste and Its Glycidyl Methacrylate Grafting and Allylation: Synthesis, Characterization and Adsorption Properties

**DOI:** 10.3390/nano11020476

**Published:** 2021-02-13

**Authors:** Elena Vismara, Giulia Bertolini, Chiara Bongio, Nicolò Massironi, Marco Zarattini, Daniele Nanni, Cesare Cosentino, Giangiacomo Torri

**Affiliations:** 1Department of Chemistry, Materials and Chemical Engineering “G. Natta”, Politecnico di Milano, 20131 Milano, Italy; giulia.bertolini@mail.polimi.it (G.B.); chiara.bongio@polimi.it (C.B.); nicolo.massironi@polimi.it (N.M.); Marco.Zarattini@polimi.it (M.Z.); 2Department of Industrial Chemistry “Toso Montanari”, Università di Bologna, Viale Risorgimento 4, 40136 Bologna, Italy; daniele.nanni@unibo.it; 3Istituto di Ricerche Chimiche e Biochimiche “G. Ronzoni”, 20133 Milano, Italy; cosentino@ronzoni.it (C.C.); torri@ronzoni.it (G.T.)

**Keywords:** cotton waste, environment, nanocellulose, adsorption, release, antibiotics, glycidyl methacrylate, wastewater, water remediation, antibacterial nanocellulose

## Abstract

Nanocellulose (NC) is getting ahead as a renewable, biodegradable and biocompatible biomaterial. The NCs for this study were recovered from industrial cotton waste (CFT) by acid hydrolysis (HNC) and by 2,2,6,6-tetramethylpiperidine-1-oxyl (TEMPO) mediated oxidation (ONC). They were functionalized by radical based glycidyl methacrylate (GMA) grafting providing crystalline HNC-GMA and ONC-GMA, and by allylation (ALL) providing amorphous HNC-ALL and ONC-ALL. HNC, ONC and their derivatives were chemically and morphologically characterized. Crystalline NCs were found capable to adsorb, from diluted water solution (2 × 10^−3^ M), the antibiotics vancomycin (VC), ciprofloxacin (CP), amoxicillin (AM) and the disinfectant chlorhexidine (CHX), while amorphous NCs did not show any significant adsorption properties. Adsorption capability was quantified by measuring the concentration change in function of the contact time. The adsorption kinetics follow the pseudo-second order model and show complex adsorption mechanisms investigated by an intraparticle diffusion model and interpreted by structure-property relationships. ONC and ONC-GMA loaded with VC, and HNC and HNC-GMA loaded with CP were not colonized by *Staphylococcus aureus* and by *Klebsiella pneumonia* and suggested long lasting release capability. Our results can envisage developing CFT derived NCs for environmental applications (water remediation) and for biomedical applications (antibacterial NC). Among the future developments, it could also be of interest to take advantage of acidic, glycidyl and allyl groups’ reactivity to provide other NCs from the NC object of this study.

## 1. Introduction

Scientists envisage nanocellulose (NC) as one of the most promising green materials due to its intrinsic properties, renewability, and abundance [1,2]. The association of high surface area with mechanical properties such as high modulus and tensile strength justifies the interest in many NC applications, not only as it is but also as a nanocomponent of composite materials [3].

NC human hazard potential literature on in vivo and in vitro tests has been reviewed by focusing on cellulose nanoparticles and nanofibers functionalized or not [4]. Even though a large part of in vitro tests shows no cytotoxicity, regarding in vivo testing there are still significant uncertainties remaining due to the scarcity of the studies that do not allow reaching the rational of a satisfying structure-property relationship. The authors suggest managing NC development with prudence, waiting for higher quality and a larger number of studies. For example, more rigorous physico-chemical characterization is needed to find and eventually avoid interference and impurity effects as NCs come from quite different sources.

Mechanical, chemical, and enzymatic treatments, or a combination of these, can be used to extract NC from natural sources [1]. In advance, the recovery from waste biomass is a concrete opportunity to take advantage of cellulose-containing waste and to enhance it [5]. NC has been mainly recovered from wood biomass, but over the past 15 years great attention has been dedicated to non-wood biomass and industrial waste [6]. Non-wood sources, including agricultural residues and industrial waste, offer an attractive alternative due to their abundance, fast generation, and low starting value. Among industrial waste, textile waste is surely a very interesting NC source. A large spectrum of cotton waste has been reduced to nano size by optimized acid hydrolysis conditions, following alkali and bleaching pre-treatments to eliminate hemicellulose, lignin, and other amorphous contents [7]. A trivial but relevant consideration is that the richer the starting material with respect to cellulose, the less important the pre-treatment. Indeed, no pulping process has been necessary before the extraction of NC from cotton linters by acid hydrolysis [8], and no chemical pre-treatment on raw cotton linter either [9].

In this work, a high-grade pure industrial cotton waste fiber, too short to be spun, was used as an NC source. Years ago, it was used to produce cotton wool, but the competition with Asian producers makes this possibility past its time. Cotton waste, namely CFT, was oxidized to ONC by means of 2,2,6,6-tetramethylpiperidine-1-oxyl (TEMPO)-mediated oxidation and hydrolyzed to HNC by acid hydrolysis. TEMPO-mediated oxidation and acid hydrolysis have been frequently exploited to provide NC [10]. ONC is a charged NC where a certain number of the -CH_2_OH groups are selectively oxidized to the –COOH group, while HNC is a quite neutral NC. An exhaustive overview of NC chemical modifications has been published, which puts into evidence practically all the features that make NCs fascinating building blocks for the design of new biomaterials [11]. In our study, ONC and HNC were modified by glycidyl methacrylate (GMA) grafting and allylation. The GMA choice was justified by our previous results that stated that GMA appendages endow cellulose with peculiar adsorption capability towards aromatic molecules and drugs [12]. We could argue that the GMA network makes GMA-cellulose a sort of ground-breaking nanostructured material. In advance, the GMA grafting of bacterial NC has enlarged its drug delivery applications [13]. NC allylation took inspiration from an efficient method of preparing allyl cellulose, and it is another approach to introduce branches on the linear cellulose polymer [14].

A question arises now: how can we enhance CFT by providing NCs and by modifying them? What kind of applications can we imagine for them? Additionally, taking inspiration from our experience of cellulose adsorption, we decided to face two apparently different problems: the removal of pharmaceuticals from water and drug release. As the removal of pharmaceuticals and drug release can be rationally considered two faces of the same medal, it could be fruitful to pursue the application of the same NCs in these two actions.

For our study, we decided to test different structures belonging to antibiotic class and to disinfectants that have in common a large diffusion in our life. The representative antibiotics, amoxicillin (AM), vancomycin (VC) and ciprofloxacin (CP), and the common antibacterial disinfectant chlorhexidine (CHX) are shown in Figure 1. Now, we are entering the core of our work. We examined if the absorption properties of these target molecules benefit from the reduction at nanoscale dimensions of cellulose, and if there are any specific chemical interactions between them and NCs.

AM, a very common wide-spectrum β-lactam antibiotic that inhibits bacterial cell wall synthesis, is administered to patients either orally or topically. As at least 50% of AM is not retained by the body, the potential for removal from environmental aqueous samples is of interest [15]. Another study concerns AM removal from aqueous solution using magnetically modified graphene nanoplatelets [16]. VC, a first-line glycopeptide antibiotic used in the treatment of infections caused by Gram-positive bacteria, such as *Staphylococcus aureus*, is always the drug of choice for antibacterial wound dressing [17,18]. CP is a second-generation fluoroquinolone, particularly effective against Gram-negative bacteria. Carboxymethylated NC has been used to fabricate a ciprofloxacin–montmorillonite composite for its sustained delivery [19]. CHX is a cationic bisguanide widely used in dentistry and cosmetic fields due to its antiseptic activity. In the literature, the controlled release and antimicrobial activity of CHX added to cellulosic substrates were investigated with microfibrillated cellulose [20] and nanocrystal membranes [21].

In the literature, it is possible to find many studies about the potentiality of NC for water remediation as well as for drug carrier systems. We selected some studies that, in our opinion, are more relevant for the aims of our work.

In the book “New Polymer Nanocomposites for Environmental Remediation” [22], a chapter has been dedicated to NCs as a sustainable material for wastewater purification. Water purification [23] is a worldwide problem, which is very difficult to solve definitively due to the large spectrum of contaminants. Important observations about NC removal mechanisms by electrostatic attraction and by surface modification based on the chemistry of hydroxyl groups open the way towards the selective adsorption of different classes of pollutants by NC. In another review, dedicated to NC-based materials, the concept of specific adsorption was extensively discussed [24]. A growing concern in today’s modern world is the contamination of water resources by pharmaceuticals [25]. The complexity of pharmaceutics and their biological capability make their removal a very difficult problem to face. At the same time, low-cost adsorbents for the removal of the main pharmaceuticals found in surface water have been focused on municipal and agro industrial waste, comprising cellulose containing waste, as a precursor, to evaluate the real applicability in water and wastewater treatment, such as competition, recyclability, and production cost. A critical review of cellulose-based nanomaterials for water purification in industrial processes cites NC for removing pharmaceuticals in the function of different NC modifications, such as sulfation and hydrophobization, that divert NC affinity towards specific pharmaceuticals [26].

Recent advances in using nanoengineering cellulose crystals and fibrils alone to develop vehicles, encompassing colloidal nanoparticles, hydrogels, aerogels, films, coatings, capsules, and membranes for the delivery of a broad range of bioactive cargos, such as chemotherapeutic drugs, anti-inflammatory agents, antibacterial compounds, and probiotics, have been reviewed [27]. This impressive list emphasizes not only the peculiar NC flexibility to provide different physical forms, but also the need to have available bioactive cargos, as outlined in the review concerning NCs as antimicrobially active materials [28]. In fact, native (as produced) NC does not possess antimicrobial properties, and this can be achieved by functionalizing NC and/or incorporating antimicrobial agents.

The overview on recent advances of NC in drug delivery systems has been focused on the strategy to merge NCs’ outstanding properties and the development of NC-based products with NC modifications, which enable NCs to have a high loading and binding capacity for therapeutic agents to control the drug release mechanism well [29]. However, the authors underline in the conclusion that many of the discussed examples did not include toxicity assessments, suggesting that the way for NC applications in the field of drug delivery is still long.

## 2. Materials and Methods

### 2.1. Materials

The starting cotton waste material (CFT) was supplied by an Italian company (C.F.T. PIETRO MASSERINI S.R.L. Company, Gazzaniga, BG, Italy). Hydrogen peroxide (H_2_O_2_) solution 30% (*w/w*), glycidyl methacrylate (GMA), ferrous sulfate heptahydrate (FeSO_4_∙7H_2_O), 2,2,6,6-tetramethylpiperidine-1-oxyl radical (TEMPO), sodium chloride (NaCl), potassium bromide (KBr), tetra-n-butylammonium fluoride (TBAF), allyl chloride, potassium dihydrogen phosphate (KH_2_PO_4_), phosphoric acid (H_3_PO_4_), sodium hydroxide (NaOH), sulfuric acid (H_2_SO_4_, 95.0–97.0% ACS grade), sodium hypochlorite solution (NaClO, 10%), chlorhexidine digluconate (20% in H_2_O), hydrochloric acid (HCl, 36.0–38.0% in water), dimethyl sulphoxide (DMSO), absolute ethanol (EtOH), methanol (MeOH), ciprofloxacin (≥98.0%), triethylamine (≥99%, TEA), n-hexane, water and acetonitrile (≥99.9%) for HPLC were purchased from Merck Life Science S.r.l. Milano Italy. Amoxicillin trihydrate and vancomycin Hikma were purchased from dott. Ambreck Pharmacy (Milano, Italy).

### 2.2. Synthetic Procedures

#### 2.2.1. HNC Preparation

An aqueous solution of H_2_SO_4_ 64% (*w*/*w*, 130 mL) was heated to 45 °C in a flask with an oil bath. Milled CFT cellulosic fibers (4.0 g) were slowly added to the acidic solution under vigorous stirring. The mixture was maintained under constant magnetic stirring at 45 °C for 6 h. Then, the reaction was quenched by pouring cold water (700 mL, T = 10 °C) into the obtained brownish mixture and the precipitation occurred in 24 h, favored by cold temperature. The supernatant was removed from the nanofibers by decantation. The remaining supernatant and the excess of reagents were removed by centrifugation at 4000 rpm (1600× *g*) for 15 min. The recovered pellet was washed by the addition of water and further centrifuged (4000 rpm for 15 min). Finally, the viscous pellet was dialyzed against distilled water (15 times the volume of dialysate; molecular weight cut off: 3500 Da). The dialysate was replaced three times with fresh water, until its pH did not decrease anymore. After dialysis, HNC suspension was obtained from the remaining brownish residuals by four centrifugations at 8000 rpm (6500× *g*) for 25 min. This procedure provided a nearly quantitative recovery of HNC (3.7 g).

#### 2.2.2. ONC Preparation

An aqueous mixture of TEMPO reagent (75 mg) and potassium bromide (KBr, 571 g) was prepared and sonicated for 10 min (215 mL). Milled CFT cellulosic fibers (4.0 g) were soaked into TEMPO/KBr solution for 1 h. Then, NaClO 10% solution (17.4 mL) was added and the reaction proceeded under stirring, maintaining the pH between 10 and 11 with drops of NaOH solution (0.5 M). When no more variations of the pH occurred, the reaction was considered over. The reaction was quenched with EtOH (20 mL). The reaction was acidified by HCl. ONC was recovered partly by precipitation and partly by centrifugation at 4000 rpm for 30 min. The collected ONC was exhaustively washed and suspended in basic water (pH = 11) by sonication with an immersed probe at 0 °C, with an output power of:20% for 30 min (with a stop of a few minutes at half time)40% for 30 min (with a stop of a few minutes at half time)

After 1 h of sonication, the slurry became transparent. This procedure provided a nearly quantitative recovery of ONC (3.8 g).

#### 2.2.3. GMA Grafting. HNC-GMA and ONC-GMA Preparation

The synthetic route was the same for HNC and ONC. NCs on a 50–700 mg scale were soaked in water (preheated at 80 °C) for 30 min. Later, FeSO_4_∙7H_2_O and H_2_O_2_ (30% *w/w*) were added and the suspension was stirred for 25 min at 80 °C. After 25 min, GMA was dropped into the reaction mixture, and it was allowed to react for 15 min (80 °C). The reagent amounts are reported in Table 1.

The reaction was allowed to cool at room temperature. The reaction showed a solid phase that was recovered by centrifugation at 6000 rpm for 15 min. The solid was further centrifuged two times with water at 6000 rpm (3500× *g*) for 15 min to completely remove the reagents excess. ONC-GMA and HNC-GMA were lyophylized as nanofibrillar powders or were obtained as suspension by centrifugation at 12,000 rpm (14,000× *g*) for 15 min.

#### 2.2.4. Allylation. HNC-ALL and ONC-ALL Preparation

The synthetic route employed was the same for HNC and ONC. NCs (348 mg) were suspended and stirred in 30 mL of DMSO under an argon inert atmosphere. After 10 min, 4 g of TBAF was added. The temperature was raised to 60 °C and the reaction proceeded for 1 h, giving a yellowish color to the reaction mixture. At 60 °C, NaOH powder (7.3 g) was added, followed by a further 50 mL of DMSO. The system was maintained in the dark for 30 min. Then, allyl chloride (18 mL) was added, and the reaction was maintained at 50 °C for 48 h. Herein, the color of the mixture turned from yellow to brown. MeOH (100 mL) was added and the precipitate was recovered. Finally, the solid was washed with further aliquots of MeOH by means of centrifugation at 6000 rpm for 15 min. The reaction product was extracted in Soxhlet with *n*-hexane for 24 h.

### 2.3. ONC Titration

The ONC potentiometric titration was performed using a NaOH solution (0.1 M). A 0.5% suspension of ONC was used. The degree of oxidation (DO) is calculated as the ratio of the added moles of NaOH and the moles of the ONC in suspension.

### 2.4. NC Loading Experiments

Aqueous solutions of AM, VC, CP and CHX were prepared (2 × 10^−3^ M). NC powder in the range of 20–100 mg was put into contact in a flask with 25 mL of each prepared solution and continually shaken in a thermostatic bath (Julabo SW22, 100 rpm, 25 °C) for 72 h. The sequestration of active ingredients promoted by NC powders over time was quantified by direct UV acquisition or HPLC injection of collected aliquots. The absorbance values of AM (λ_max_ = 272 nm), VC (λ_max_ = 280 nm), CP (λ_max_ = 229 nm) and CHX (λ_max_ = 255 nm) were converted in concentration by using previously created calibration curves.

#### 2.4.1. UV Spectroscopy

The amount of adsorbed/released molecules over time was monitored through quantitative UV–VIS analysis (UV-spectrophotometer JASCO V-650 and SpectraManager software version 1; Jasco Corporation, Cremella, LC, Italy). Several aliquots (100 µL) of HNC and ONC adsorption solutions (AM and VC) were collected at different times and submitted to UV acquisition after a volume made up to 5 mL. The acquisition wavelength range was set from 500 to 240 nm. At least two different sampling and acquisition procedures for each point were performed.

#### 2.4.2. HPLC Methods

The column is a Hypersil BDS C18 (250 × 4.6 mm, 5 µm) on a Knauer Smartline Pump 1000 equipped with autosampler LC, 6-Port/3-Channel Injection Valve, vacuum degasser, and fraction collector (Knauer, Berlin, Germany) 

CP: the mobile phase is composed by acetonitrile:water:triethylamine (20:80:0.3 *v/v/v*). The pH was adjusted at 3.3 with H_3_PO_4_. The UV-detection wavelength is set at λ = 229 nm and CP eluted in 5 min setting a flow = 1 mL/min. Each injection (20 µl) was performed twice at room temperature. The linear range of concentrations used for the HPLC calibration curve (y = 157.63x + 525 with R^2^ = 0.998) is from 10 mg/L to 100 mg/L (10, 25, 40, 50 and 100 mg/L). The different dilutions were obtained from a CP stock solution (200 mg/L) in mobile phase.

CHX: the mobile phase is composed by buffer: acetonitrile (68:32, *v/v*), being the buffer of a 0.05 M solution of KH_2_PO_4_, with 0.2% of TEA and pH = 3 (adjusted with H_3_PO_4_). CHX is detected at λ = 255 nm and it has a retention time of 8.8 min, with a flow of 1 mL/min. Each injection (20 µL) was performed twice at room temperature. The linear interval of concentrations used for the HPLC calibration curve (y = 63.107x + 341.56 with R^2^ = 0.994) ranges from 50 mg/L to 250 mg/L (50, 100, 150, 200 and 250 mg/L). Different dilutions were obtained from a CHX stock solution (1000 mg/L) in the mobile phase.

### 2.5. Characterization Techniques

#### 2.5.1. Infrared Spectroscopy (IR)

The solid phase FT-IR spectra of the powdered sample with infrared-grade KBr were generated using an ALPHA spectrometer (Bruker, Bremen, Germany). Data were analyzed using OPUS software, version 7.0 (Bruker, Bremen, Germany). The acquisition of the spectra was performed in the range of 4000–400 cm^−1^. The estimation of GMA molar substitution ratio (MS) was obtained with the method used in our previous works, see Equation (1) [12]:(1)$${\mathrm{MS}}_{\mathrm{FT-IR}}=\frac{area\;ester\;\left(manual\;band\;integration\right)}{area\;cellulose\;\left(range\;780-465\;cm^{-1\;}integration\right)}$$

At least two tests were carried out to assure better reproducibility and accuracy for the integration calculations. The IR spectra reported in this paper are the most representative among the ones obtained. The DO of ONC was calculated from the FT-IR spectrum as well. DO corresponds to the ratio between the integrated area under the peak of the COOH group (from 1830 cm^−1^ to 1670 cm^−1^) and the characteristic peak of cellulose (from 780 cm^−1^ to 480 cm^−1^).

#### 2.5.2. Solid-State ^13^C CP-MAS NMR

The ^13^C NMR analysis was performed by using Dipolar Decoupling Cross Polarization Magic Angle Spinning (CP-MAS) technique with a ^13^C 75.47 MHz Bruker Avance 300 spectrometer (for Bruker S.r.l., Milan, Italy). Concerning data acquisition parameters, the repetition time (D1) was equal to 8 s, while the contact time and the spin rate were 1.6 ms and 10,000 Hz, respectively. To obtain good quality spectra, 2K scans were collected. The samples were positioned in a zirconium rotor (diameter: 4 mm; height: 21 mm). Tetramethylsilane was the reference substance for the chemical shifts. Benzene was used as a secondary reference standard. The crystallinity index (CrI%) of nanocellulosic materials was evaluated by means of the following equation used in our previous works, see Equation (2) [12]:(2)$$Cr.I\;\left(\%\right)=\frac{A}{A+B}*100$$
where *A* corresponds to integrals of the C4 peaks at 86–92 ppm (crystalline) and *B* corresponds to the integrals of the C4 peaks at 80–86 ppm (other components).

#### 2.5.3. Scanning and Transmission Electron Microscopy (SEM and TEM)

SEM: the surface morphology of the samples was observed by using a ZEISS EVO-50 EP (ZEISS Microscopy, Jena, Germany). This instrument can work at variable pressures (100–120 bar in this case), so the gold coating step could be avoided with a negligible loss of resolution.

TEM: transmission electron microscopy imaging was carried out by using a Philips CM 200 instrument (Philips S.p.A., Milan, Italy) at 200 KV voltage. The samples were prepared by depositing a drop of solution on a copper grid (200 mesh) coated with a thin layer of carbon and then left to dry for a few hours before the observation.

#### 2.5.4. Dynamic Light Scattering (DLS)

Hydrodynamic diameter and zeta potential (ζ) values of NC suspensions (0.005%) were measured using the Zetasizer Nano ZS (Malvern, Worcestershire, UK) with a fixed 173° scattering angle and a 633-nm-helium-neon-laser. Data were analyzed using Zetasizer software, version 7.11 (Malvern, Worcestershire, UK). The temperature was set at 298 K.

### 2.6. Antibacterial Activity Test

The tests were run according to UNIEN ISO 20645:2005 by Centro Tessile Cotoniero & Abbigliamento S.p.A. (www.centrocot.it). *Staphylococcus aureus* (ATCC 6538 LOT: DSM 799-0415) and *Klebsiella pneumoniae* (ATCC 4352 LOT: DSM 789-0513) were cultured in a Triptone and Soya medium for 24 h at 37 °C. The obtained suspensions were diluted to 108 UFC/mL. Then, 1 mL samples (108 UFC/mL) were added to 150 mL of Triptone Soya Agar (LOT: Oxoid 1837431) at 45 °C to afford culture medium. An NC sample of size 25 ± 5 mm was incubated in 5 mL of the culture medium for 24 h at 37 °C. Each test was performed in quadruplicate in Triptone Soya Agar culture medium. The reference material was 100% cotton fabric according to ISO 105/F02:2009.

## 3. Results and Discussion

### 3.1. NC Preparation

CFT is the waste cotton short-staple fibers depicted in Figure 2. The raw short-staple fibers Figure 2A were transformed by consecutive mechanical treatments into Figure 2B,C by CFT company. The material Figure 2C was used as NC precursors.

The CFT fibrous morphology was captured by an SEM image (Figure 3), which defines the actual dimensions of the micrometric cellulose fibers, in addition to their shape. The CFT SEM image is equal to untreated cotton fiber SEM images reported in the literature.

Figure 4 displays the CFT solid-state CP-MAS spectrum with the integrals. The spectrum puts into evidence the quite high purity of the sample and the crystallinity index of CFT (CrI = 62%), calculated as detailed in Section 2.5.2. 

Figure 5 reports the CFT FT-IR and ATR FT-IR spectra that can be superimposed with the regular cotton spectra. While ATR mode allows the chemical analysis of the surface and it does not need any preparation of the sample, the FT-IR transmission analyzes the bulk on the powdered sample in KBr.

CFT potentiality as an NC source was investigated by using two different top-down approaches: acid hydrolysis and TEMPO-mediated oxidation, which gave two different types of NC (HNC and ONC, respectively). The reactions were performed in a one-pot procedure directly on CFT, since the established high purity of the material allows us to avoid any pre-treatment.

#### 3.1.1. HNC Preparation

The hydrolysis of glycosidic bonds of starting cellulose chains promoted by sulfuric acid (H_2_SO_4_) breaks the cellulose fibers, providing a white powder. Consequently, CFT morphology underwent a microscopic alteration with respect to CFT, as depicted by SEM and TEM images shown in Figure 6.

SEM and TEM microscopies furnish details about the changes in the materials. While the SEM image depicts HNC assembled in well separate chips of different sizes and shapes, TEM shows the nanoscale dimensions of the distinguishable HNC rods. Herein, HNC rods display dimensions of 100–300 nm in length and around 5–20 nm in width.

Concerning the structural characterization, as expected, the comparison between IR spectra of CFT and HNC does not show any other group introduced on the cellulose chain (see Appendix A). However, there is a difference in the fingerprint C-O-C region between 1100 and 1250 cm^−1^ that could be ascribed to the different C-O-C induced by the CFT depolymerization/degradation to the nano size. The non-relevance of signals characteristic of sulfate groups suggests that the sulfation degree is very low. This was confirmed by the HNC titration run according to Section 2.3 that gave a flat curve.

The solid-state CP-MAS NMR spectrum in Figure 7 permitted a more detailed investigation of the structural features of HNC. All the detected signals are typical of pure cellulose. The CP-MAS NMR technique was also exploited for an accurate measurement of crystallinity index (CrI%), 71%, as detailed in Section 2.5.2.

In the HNC cellulose NMR spectra reported in Figure 7A, the red trace is related to the components of the rigid molecular structure (CP-MAS experiment), while the high power decoupling magic angle spin (HP-DEC MAS), blue trace, is related to the high mobility components The difference spectrum (green trace) gives an idea of the morphological distribution between rigid and mobile components (signal intensity normalized to signal noise). Figure 7B reports in the central chain the scheme of the hydrogen bonds of the beta sheets, which are the basis of the crystalline form of cellulose. The surface chains as well as the folds induce greater freedom to the glucose residues and primary alcohol derivatizations are possible, as in the case of treatment with sulfuric acid or the oxidation process.

#### 3.1.2. ONC Preparation

The second followed top-down approach for the recovery of NC from CFT is TEMPO-mediated oxidation. ONC is a charged cellulose where the primary hydroxyl groups are selectively oxidized to COOH. Similarly to what happened with HNC, ONC loses the aspect of the fibrous and compact starting material, as shown by SEM and TEM images reported in Figure 8.

SEM on the left refers to lyophilized ONC in the form of aerogel. On the right, TEM validates the definition of ONC as cellulose nanofibrils (NFC), since the filaments have a length and width of about 100–200 nm and 10–20 nm, respectively.

Two ONC FT-IR spectra are displayed in Figure 9, depending on the COOH group, in the acidic form (blue) or in the salted form (red).

The characteristic band of carboxylate anion is around 1600 cm^−1^, overlapping with the signal of bound water of cellulose (red). The band at 1729 cm^−1^ is descriptive of the COOH group (blue). In advance, it is possible to observe a difference in intensity in the OH bond (3500–3000 cm^−1^) region between ONC in its acidic (blue) form and salted (red) form due to the increase in the COOH groups. Other bands representative of the COOH group (between 1440 and 1375 cm^−1^) are negligible due to overlapping with other stronger characteristic signals of cellulose. The area of the COOH band can be used to calculate the degree of oxidation (DO), equal to 0.32, as detailed in Section 2.5.1. ONC was also titrated by potentiometric measures, as detailed in Section 2.3. DO was obtained by the ratio of the moles of NaOH used for the titration with the moles of the ONC in suspension (see Appendix A). ONC’s DO by titration is 0.39, quite similar to the DO obtained with FT-IR.

Figure 10 reports the ONC ^13^C CP-MAS NMR spectrum; the new signal at 177 ppm is attributed to the quaternary carbon of the introduced carboxyl group of ONC. The CH_2_OH oxidation is partial, as confirmed by the presence of C_6_ signals. ONC CrI% is equal to 65%.

### 3.2. GMA Grafting

GMA grafting was performed in the presence of Fenton’s type reagent, according to the approach previously developed for cellulose [12]. The GMA grafting radical-based mechanism is detailed in Scheme 1.

GMA grafting was performed by a one-pot methodology, whose first step is cellulose activation by generating carbon-centered radical intermediates with a Fenton-type reaction, and the second step is radical quenching by the GMA double bond forming a new stable C-C bond between cellulose and GMA. The carbon-centered radicals are formed by hydrogen abstraction in any position of the glucose ring due to the high reactivity of the OH radical and are quenched by the GMA double bond. For convenience, the C3 radical formation is reported. Thus, a statistical distribution of GMA appendages occurs for each glucose unit in terms of numbers and position, and it is quantified by the average GMA appendages molar substitution degree (MS). MS was measured by FT-IR as detailed in Section 2.5.1. MS can be measured by gravimetric analysis, too. As already observed with regular cellulose, the quantitative analysis by FT-IR spectroscopy is in good agreement with the GMA grafting gravimetric quantification. A great effort was made in order to find a close correlation between molar ratios of Fenton’s reagent/GMA and MS suitable for desirable applications. GMA content is not the only parameter that influences MS. The amounts of catalyst (FeSO_4_∙7H_2_O) and oxidizer (H_2_O_2_) significantly influence the MS as well. Several functionalization reactions with different ratios of reagents were performed, and they are summarized in Table 1 (see Section 2.2.3). MS higher than 2.0 corresponds to materials that are not homogeneous where the NC core is covered by granules of GMA (see Appendix A).

Entries 1–4 on the 50 mg ONC/HNC scale show the MS trend by decreasing the reagent amount. ONC and HNC behave in the same way. The same preparation was repeated at least five times to test the reproducibility and the MS range that were found acceptable. Entries 5–8 report the scale-up from 50 to 700 mg. As expected, MS decreases for the same reagent ratios. NMR characterization, microscopy, and suspension DLS and zeta potential were run on GMA functionalization in the range of 0.3–0.8 MS, while adsorption investigation was performed on a larger range (see Section 3.6).

#### 3.2.1. HNC-GMA

The FT-IR spectrum of HNC-GMA (green) is reported in Figure 11 and compared with the starting HNC (blue). It contains the bands of the regular cellulose spectrum, with the addition of the glycidyl ester band at 1722 cm^−1^, thus confirming the occurred GMA grafting [12]. Appendix A is representative of two different MS for HNC-GMA.

Concerning the morphology, the SEM image in Figure 12 suggests that HNC morphology did not substantially change once grafted with GMA.

A slight difference can be identified in the texture of the surface; the HNC surface is smoother than the HNC-GMA one. TEM images were useful to evaluate the average dimensions of the HNC-GMA nanofibrils. Similarly to the other samples, the shape was found to be a rod-like shape. The functionalization does not affect the shape of the NC, while the dimensions of HNC-GMA (around 960 and 83 nm in length and width, respectively) are different with respect to the dimensions of HNC. This variation can be due to the coverage with GMA of the nanofibrils that leads to an increment in the width.

#### 3.2.2. ONC-GMA

The FT-IR spectrum of ONC-GMA (purple) is reported in Appendix A. Because of the overlapping of COOR and COOH, it is necessary to transform COOH into COONa to identify the glycidyl ester band (see in the figure the starting ONC in the salted form (blue)). FT-IR grafting quantitative analysis was performed according to Section 2.5.1.

The grafting on ONC was also confirmed by the presence of GMA signals in the ^13^C-CP-MAS NMR spectrum (Figure 13). There is no presence of signals related to olefinic carbons, which should be located in the range 120–160 ppm between the signal of the carbon C1 of the cellulose (~105 ppm) and the signal of the C=O of the ester group (~178 ppm). This confirms the absence of unreacted GMA. Figure 13 also reports the FT-IR spectrum. MS was 0.21 by FT-IR and 0.21 by ^13^C-CP-MAS NMR, in perfect agreement.

The microscopic images (SEM and TEM) of ONC-GMA are shown in Figure 14. The estimated dimensions are of about 780 nm in length and 82 nm in width, outlining an increment in comparison with dimensions of the native ONC.

### 3.3. Allylation

Scheme 2 shows the NC decorated with allyl appendages, performed according to Section 2.2.4. Contrary to GMA grafting, allylation interests the OH groups. As concerns HNC, there are three OH suitable to substitution, while for ONC the partial oxidation in six C6 reduces the number of the OH.

Figure 15 and Appendix A show the ONC-ALL and HNC-ALL^13^C CP-MAS NMR spectra, respectively. The assignment of the signal is reported for the ONC-ALL spectrum in Figure 15. This type of cellulose functionalization highly affects the crystallinity. The typical profile of cellulose allomorph I disappears. More in detail, C4 (crystalline) around 89.5 ppm practically disappears, and the C3All is shifted downfield by about 8 ppm, overlapping the C4 signal, while the C6All shifts towards low fields, allowing its integration. The remaining cellulose signals shift at 78.8 ppm, together with OCH_2_All signals.

The assignments of the signals have been made on a correlative basis and are shown in the figure. The comparison with the starting ONC spectrum (Figure 15C) highlights the loss of any form of crystallinity, while the feeble response of the anomeric signal of cellulose in the HP-DEC spectrum (Figure 15B), which substantially records the mobile components of the polymer, indicates the maintenance of a certain rigidity of the polymer chain. A degree of substitution (DS) of 2.9 is obtained from the CP-MAS spectrum by the ratio between the average value of the areas of the allyl signals, at 118 and 135 ppm, and the NC anomeric signal at 100 ppm (data not shown).

Table 2 reports the DS calculated by the ratio between the average integration of allyl signals and the anomeric signal integration. A tentative evaluation of the distribution of the allyl substituent on the ONC glucose unit, reported as entry 2 in Table 2, is measured for C6_All_ by direct integration, for C3_All_ by subtracting the anomeric integration from the area of C4 + C3A_all_, and for C2_All_ by the difference with respect to DS.

Appendix A reports the HNC-ALL and ONC-ALL ATR FT-IR spectra compared with the HNC and ONC precursors, respectively. In agreement with NMR DS data, it is interesting to note the disappearance for HNC-ALL and the decrease for ONC-ALL of the OH peaks between 3500 and 3000 cm^−1^. For ONC-ALL, -COOH still remains.

### 3.4. NC Crystallinity

Table 3 summarizes the CrI% of the recovered ONC and HNC and their grafted derivatives, compared with the CrI% of CFT. They were all calculated from solid-state ^13^C CP-MAS NMR spectra [30]. The NMR method measures a lower crystalline index than the XRD method, which underestimates the amorphous content of cellulose [31]. The use of the NMR method was further supported when NC has been extracted from citrus waste [32].

As regards ONC and HNC, there are small but significant variations with respect to the starting CFT. Generally speaking, the preparation of NC acts on the amorphous part of the cellulose, thus increasing the CrI of the NC with respect to the starting material. This is true for HNC, which passes from 62% of the CFT to 71% of HNC. The same can be said for ONC, but the increment for ONC is very low—from 62 to 65. In our opinion, as the oxidation process only regards position 6, it is partial, it is not a degradative process, and it does not influence so much the ratio between the amorphous and the crystalline structure. ONC and HNC values of CrI% were compared with the ones of HNC-GMA and ONC-GMA derivatives. The CrI% decreases for both. Since the GMA grafting acts on the C–H group (see Scheme 1), the OH groups are not interested by the radical activation. GMA grafting partially hampers the strong polar interactions between the different cellulose chains that are responsible for the ordered cellulose structure.

Contrary to GMA grafting, allylation interests the OH groups, like acetylation. This type of cellulose functionalization highly affects the crystallinity indeed. The adopted reaction conditions fully allylated both HNC and ONC. Due to the high DS, see Section 3.3, HNC-ALL and ONC-ALL completely lost their crystalline nature, as reported in Table 3, and they are no longer crystalline NCs.

### 3.5. NC Suspensions DLS and Zeta Potential (ζ)

The dimensions found with the DLS technique (Table 4 and Appendix A) are related to the average of the three dimensions of the nanofibrils of the NC. The results show values on the nanometer scale both for starting and GMA functionalized NCs, as anticipated by microscopies.

The native cellulose nano fibrils ζ at pH = 7 decreases from −35mV to −65mV by TEMPO oxidation [33]. As expected, TEMPO oxidation makes NC ζ more negative [34]. For our samples, the charged ONC ζ is more negative, while the almost quite neutral HNC is less negative. Functionalization with GMA makes ONC-GMA ζ less negative. The steric hindrance of the GMA networks may provoke a decrease at the level of the surface charge of functional groups of ONC. As expected, HNC and HNC-GMA ζ are the same.

### 3.6. Adsorption Experiments on ONC, HNC and Their GMA

The adsorption capability of native and functionalized NCs was tested with AM, VC, CP and CHX from 2 × 10^−3^ M aqueous solution at 293 K. The adsorption process is detailed in Section 2.4. The amount of the adsorbed molecules by different NCs was measured as a function of the contact time, and the curves of adsorbed amount (mg/g NC) versus time (min) are reported in Figure 16 and Figure 17. Table 5 reports the adsorbed amount at 24 h, which is the reference time for the equilibrium, as discussed below. The allylated NC adsorption process gave very poor results which are not reported.

Table 6 reports the removal capability from the water solution (2 × 10^−3^ M), expressed in percentage.

If the bar is fixed on our choice at 50%, every NP might be developed for at least one specific removal. Furthermore, ONC might remove AM, VC and CHX at the same time. Only CP% removal is far from 50% for all the NCs.

### 3.7. Kinetic Studies

From Figure 16 and Figure 17, it is evident that the adsorptions involve many stages. Noteworthy, the same trend has been observed for AM using magnetically modified graphene nanoplatelets (M-GNPs), where the AM adsorption occurred in two stages [16].

Kinetic models have been specifically investigated for the sorption of pollutants from aqueous solution, and the pseudo-second order kinetic model reported in Equation (3) has been selected [35].
(3)$$\frac{t}{q_{t}}=\frac{1}{k_{2}q_{e}^{2}}+\frac{t}{q_{e}}$$

The experimental data in Figure 16 and Figure 17 were elaborated according to this model, and the calculations are reported in Table 7, where q_e_ and q_t_ are the amount of target molecule adsorbed (mg/g NC) at equilibrium and at time t (min), and k_2_ (g mg^−1^ min^−1^) are the rate constant of second order adsorption. The rate constant range for the pseudo second order model was found to be 10^−3^–10^−5^.

The intraparticle diffusion model was also employed to study the diffusion mechanism for adsorbate/adsorbent interaction, see Equation (4) [36]:(4)$$q_{t}=k_{d}t^{1/2}+C$$
where k_d_ is the intraparticle diffusion rate constant.

Many examples of the intraparticle diffusion model’s use have been found in the literature. Weber and Morris suggested this kinetic model to identify the diffusion mechanism and rate controlling step that affect the adsorption process. According to this model, the plot of q_t_ versus t^1/2^ can give a straight line, where k_d_ can be calculated from the slope. For this model, if the plot q_t_ versus t^1/2^ passes though the origin, the intraparticle diffusion is the only rate limiting step. The deviation from the origin suggests that although intraparticle diffusion was involved in the adsorption process, it was not the sole rate-controlling step. Amino modified NC adsorbent for boron was studied according to the Weber–Morris model [37]. Three steps were found and explained by film diffusion, particle diffusion and internal surface adsorption. In advance, the sodium itaconate grafted NC, used for the elimination of lead ions from water, shows a three staged chemisorption process by external adsorption, intraparticle diffusion and chemisorption equilibrium [38]. For discussing the NC interaction, it is also convenient to cite the kinetic studies of fluoride adsorption onto activated alumina, alum and brick powder, where the adsorption kinetics involve film diffusion and intraparticle diffusion [39].

Figure 18 reports the use of the intraparticle diffusion model for VC/ONC (Figure 18A), CHX/HNC (Figure 18B), CHX/ONC (Figure 18C) and CHX/ONC-GMA (Figure 18D). For the other target molecules/NCs, the intraparticle diffusion model did not fit.

We discuss our kinetic studies according to the pseudo second order model and intraparticle diffusion model results. MolView (Open-Source chemical modeling package, https://molview.org/) helps us to visualize molecular structures and view their properties (see Appendix A).

Amoxicillin in Appendix A appears as an angular polar molecule with two not hindered reactive groups, an amino and a carboxylic group. In Table 7, entries 1 and 2, the pseudo second order model shows that the ONC rate constant is higher than the ONC-GMA rate constant, while the q_e_ are quite the same. The AM amino group can interact with the ONC carboxylic group, while the AM carboxylic group can interact with the hydrophilic ONC surface. AM was adsorbed by ONC in large amounts due to strong and specific interactions between AM and ONC. Since COO^−^ is not involved in the GMA grafting reaction, no interference with the AM affinity towards ONC should be seen. In fact, the GMA network does not avoid AM adsorption, but only slows down the rate by steric hindrance. As AM/ONC and ONC-GMA data did not fit with the intraparticle diffusion model, the interaction might occur mainly on the NC surface. HNC did not adsorb AM at all. Comparing AM affinity for the native ONC and HNC, ONC is a partially carboxylated NC, and from our results, it appears that this structural change dramatically enhances its adsorption capability with respect to HNC. At least with AM, HNC is nothing more than cellulose, and neither nanostructure endows HNC with adsorption capabilities with respect to regular cellulose. A further confirmation of the specific interaction between AM and ONC comes from the fact that the nonexistent affinity of HNC towards AM benefits from increasing GMA network insertion, reaching values of AM adsorption far from the percentages seen for ONC-materials (see entry 3 in Table 7). Noteworthy, the trend is that the increase in MS corresponds to an increase in adsorption (see Table 5, entry 1). The constant rate is the same as for ONC, see entry 3 in Table 7, but as already outlined, the chemical interactions are quite different.

Vancomycin in Appendix A appears as a big, flat and rigid molecule, with a primary amino group and a secondary amino group on the more flexible part of its very complex structure. Similar to AM, VC was strongly adsorbed by ONC (see entry 4 in Table 7). A good explanation can be the specific interaction between the amino groups of VC and the COOH of ONC. The rate constant is of the order of magnitude of 10^−5^, figured out in a slow process. Figure 18A shows the diffusion model. The deviation from the origin suggests that although intraparticle diffusion was involved in the adsorption process, it was not the sole rate-controlling step. Actually, for VC the diffusion model shows two steps that might be the external adsorption on the ONC surface (fast as shown in the figure), followed by the intraparticle diffusion (slow as shown in the figure). Contrary to AM, VC adsorption was negatively affected by the medium MS GMA grafting (see entry 5 in Table 7). There is a relationship between the GMA network on the NC surface (dictated by MS values) and its capability to capture molecules of a large size and shape. The GMA network avoids VC to come into contact with the COOH group by steric hindrance. VC did not follow the intraparticle diffusion model with ONC-GMA, but the ONC-GMA rate constant is two orders of magnitude higher than the ONC rate constant (see entry 5 in Table 7). Putting together the pseudo second order model and intraparticle diffusion model, we could argue that VC diffuses slowly into ONC, while with ONC-GMA the adsorption quicky occurs, but only on the surface, giving rise to a lower adsorption. As HNC did not adsorb VC at all and the GMA effect is negative for VC, we did not test HNC-GMA with VC.

Ciprofloxacin in Appendix A appears almost planar, polar and negatively charged. The rate constants in entries 6–9 are of the order of magnitude of 10^−4^. The intraparticle diffusion model did not fit at all. As steric hindrance is not a distinctive tract of CP, being a small molecule with quite the same MW as AM, we could argue about surface interaction, but it is actually very modest. In fact, CP is negatively charged due to the carboxylic group. As expected, the CP/ONC couple gave the worst result (see entries 6 and 7 in Table 7). The network of GMA on ONC slightly enhances the absorption towards CP, in part counterbalancing the same charge repulsion (entry 8 in Table 7). In conclusion, the nanostructure itself endows HNC and ONC with a smooth CP adsorption capability and the adsorption benefits of the low grafted GMA that, with their branches, increases the nanomaterial surface area.

The behavior of chlorhexidine is, in our opinion, very intriguing. CHX in Appendix A appears positively charged, very polar and flexible, contrary to AM, VC and CP. CHX reacted very slowly with ONC and HNC, the rate constant being of the order of magnitude of 10^−5^, figured out a slow process, but it was adsorbed in a good amount (see entries 10 and 11 in Table 7). It shows a complex diffusion process with HNC in Figure 18B and with ONC in Figure 18C. The deviation from the origin suggests that although intraparticle diffusion was involved in the adsorption process, it was not the sole rate-controlling step. We could figure out its behavior by considering its high conformation mobility due to the CH_2_ chain. Even though CHX appears to be a big molecule, it might adapt itself in attaching to the external NC surface and/or in diffusing into the film that covers the NC surface due to favorable polar interactions (fast as shown in the figure, especially for HNC), then slowly entering and diffusing and finally reaching the chemisorption equilibrium. The GMA network made the process worse for HNC and ONC, but faster (see entries 12 and 13 in Table 7). As the GMA network makes the surface less hydrophilic and more rigid, it reduces the CHX interaction in both cases. However, Figure 18D shows that ONC and ONC-GMA act in the same way, suggesting a quite similar mechanism.

### 3.8. Antibacterial Activity

In vitro antibacterial tests described in Section 2.6 were performed on the two strategic drugs such as VC and CP, loaded on ONC and ONC-GMA and on HNC and HNC-GMA. The samples submitted to the antibacterial tests were found capable of recharging quite the same amount of VC and CP after an exhaustive washing procedure. Every test was repeated four times. Figure 19 reports for convenience one image for each test.

All the samples were not colonized by *Staphylococcus aureus* (*S. aureus*) and *Klebsiella pneumonia* (*K. pneumoniae*), and the presence of the inhibition zone demonstrated without a doubt that they are antibacterial nanomaterials. The inhibition zone measurements reported in Table 8 are not an absolute value, also for the irregular shape of the samples; nevertheless, they suggest some observations by comparison with the loaded drug amounts reported in Table 7.

The same inhibition zones for HNC and for HNC-GMA (entries 1 and 2) support the hypothesis that HNC-GMA holds part of the loaded CP. Similarly, the fact that for VC the inhibition zone is double for ONC compared with ONC-GMA (entries 3 and 4), being the ratio of loaded VC more than five times, suggests that ONC holds part of the loaded VC.

## 4. Conclusions

In our study, the NCs’ precursor is industrial cotton waste, which guarantees a high purity grade. This choice has an environmental meaning, too. The cotton fibers were reduced to the nano dimension by acid hydrolysis, providing HNC that is simple “cellulose in the nano form”. In addition to the cotton fiber reduction to the nano dimension, the cotton modifications provided negatively charged NC (ONC), GMA grafted NC (HNC-GMA and ONC-GMA) and allylated NC (HNC-ALL and ONC-ALL). The first goal of this study was to provide, as much as possible, detailed structural and morphological characterizations of the proposed NC structures with the aim of discussing their structure-properties relationship, including adsorption capability towards target molecules with different structures. Our results demonstrate that chemical adsorption processes occur, and that it is possible to correlate the target molecules’ chemical structures, polarity, size and shape with the NCs’ chemical structures, functionalization degree, morphology and surface hydrophobic/hydrophilic character. Actually, the capability of NCs to give rise to specific interactions managed by chemical and physico-chemical parameters is, in our opinion, a very important result of our studies. We are confident that the tested NCs’ behavior will be crucial to identify their most promising applications that could be in the environmental and biomedical fields. Among the future developments, it could also be of interest to take advantage of the reactivity of the COOH group, of the epoxide group and even of the allyl group, to provide other NCs from the NC object of this study.

## Data Availability

Not applicable.

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
