# Peer review of "Nanocellulose from Cotton Waste and Its Glycidyl Methacrylate Grafting and Allylation: Synthesis, Characterization and Adsorption Properties"

_nanomaterials, 2021, doi:10.3390/nano11020476_

Round 1
Reviewer 1 Report
This work focuses on nanocellulose (NCs) which was recovered from an industrial cotton waste (CFT) by acid hydrolysis (HNC) and by TEMPO mediated oxidation (ONC). It was found capable to adsorb from diluted water solution the antibiotics vancomycin, ciprofloxacin, amoxicillin and the disinfectant chlorhexidine while amorphous NCs did not show any significant adsorption properties. (Adsorption capability was quantified by measuring the concentration change in function of the contact time).
1. The CFT fibrous morphology was captured by SEM image and defines the actual dimensions of the micrometric cellulose fibres, in addition to their shape. I suggest the authors to move the SEM photographs from "supplementary material" to Figure 2 of the main body of the manuscript (and comment on it).
2. The authors present ATR FT-IR spectra of CFT in Figure 4. Comment/discuss about the spectra. Move the text that corresponds to FT-IR closer to the FT-IR spectra.
3. I suggest the authors to comment on the antibacterial activity of their samples. Could they re-use them?
4. Maybe some information should be moved from supplementary to the main manuscript. this back and forth in not so convenient.
5. A few typos and syntax issues should be resolved.
This paper could be published after completing the above minor issues.
Reviewer 2 Report
-
This manuscript is focused on the preparation and characterization of the nanocellulose particles modified with glycidyl methacrylate or allyl chloride. The obtained nanoparticles have also been characterized for adsorption of antibiotics (amoxicillin, vancomycin, and ciprofloxacin) and antibacterial activity of antibiotic-loaded particles. Generally, the paper is well written and seems scientifically competent within my areas of expertise. In my opinion, the manuscript can be considered for publication in Nanomaterials after addressing the following comments:
- Title: The title of the article needs to be corrected. “Synthesis, characterisation and antibacterial principles adsorption properties” sounds strange and does not quite match the paper content.
- Scheme 1: Draw the structures of cellulose in the same way as in Figure 3, avoiding angles around the glycosidic bond (the angle in the structural formula denotes a carbon atom). Correct the Fenton reaction equation so that all the components are at the same level.
- You need to improve the quality of figures 3,4,5,7,8,10,12, on which the numbers and scale bars are barely visible.
- Table 4: The standard deviation should be expressed as ONE significant figure; that is, unless the number is between 11 and 19 times some power of ten, in which case you can use two significant figures. The mean value should be rounded off at the decimal place corresponding to the last significant digit of its standard deviation. E.g., 155.8 ± 18.3 (column 3) should be presented as 156 ± 18; 140.5 ± 3.5 (column 3) should be presented as 141 ± 4, etc.
- Line 534: Do you mean mmol/g instead of mmoles?
- Table 7: Replace “exp” with “´10”.
- Table 8: Provide the standard deviations and the number of measurements.
- Across the text: replace ξ (xi) with ζ (zeta).
- Across the text: Check the appropriateness of using acronyms. Overuse or inappropriate use of acronyms makes reading paper difficult. Do not introduce an acronym unless you will use it (e.g., TBAF, line 150; MeOH, line 153). Spell out acronyms on the first use (e.g. VC and CP, line 27). Do not mix up the acronyms (e.g., HNC-All and HNC-ALL; ONC-All and ONC-ALL). Once you introduce the acronym, use it consistently thereafter (e.g., you used “nanocellulose” 44 times in the text, even though you introduced the acronym NC).
Round 2
Reviewer 2 Report
The authors have successfully addressed all my concerns, improving the manuscript with their edits. In my opinion, the manuscript is now acceptable for publication.
This manuscript is a resubmission of an earlier submission. The following is a list of the peer review reports and author responses from that submission.
Round 1
Reviewer 1 Report
Vismara and coworkers investigated nanocellulose (NC) from cotton waste and its glycidyl methacrylate grafting and allylation for drug delivery systems. The idea and motivation are reasonable, and the characterizations are well performed. However, the following issues must be addressed before it can be published in Nanomaterials.
- Line 98, ‘Nanocellulose’ should be changed to ‘NC’.
- The image resolution of the figures and the equations is too low to be published. The authors must make efforts to improve it.
- It is better to show the DLS curves not only the numbers.
- What are the sizes for unmodified NC, All-ONC and All-HNC, which should be included into Table 4?
- In table 4, what does the ‘1’ in ‘NC1’ mean?
Author Response
Reviewer 1
- Line 98 changed
- Resolution of the figure improved
- Added DLS curve in supplementary information
- Table 4: unmodified NC, ALL-ONC and ALL-HNC cannot be suspended in water
- Table 4 added footnote
thanks for your suggestion best regards
Reviewer 2 Report
This is a well written paper that should be published.
My only comment is that in some of the Electron Microscopy figures, the scale bars are either abscent or hard to read
Author Response
- Improved electron microscopy figures
Thank you for your suggestion, best regards